# Pseudo-Extended Markov Chain Monte Carlo

**Christopher Nemeth**
Department of Mathematics and Statistics
Lancaster University
United Kingdom
c.nemeth@lancaster.ac.uk

**Fredrik Lindsten**
Department of Computer and Information Science
Linköping University
Sweden
fredrik.lindsten@liu.se

**Maurizio Filippone**
Department of Data Science
EURECOM
France
maurizio.filippone@eurecom.fr

**James Hensman**
PROWLER.io
Cambridge
United Kingdom
james@prowler.io

## Abstract

Sampling from posterior distributions using Markov chain Monte Carlo (MCMC) methods can require an exhaustive number of iterations, particularly when the posterior is multi-modal as the MCMC sampler can become trapped in a local mode for a large number of iterations. In this paper, we introduce the pseudo-extended MCMC method as a simple approach for improving the mixing of the MCMC sampler for multi-modal posterior distributions. The pseudo-extended method augments the state-space of the posterior using pseudo-samples as auxiliary variables. On the extended space, the modes of the posterior are connected, which allows the MCMC sampler to easily move between well-separated posterior modes. We demonstrate that the pseudo-extended approach delivers improved MCMC sampling over the Hamiltonian Monte Carlo algorithm on multi-modal posteriors, including Boltzmann machines and models with sparsity-inducing priors.

## 1   Introduction

Markov chain Monte Carlo (MCMC) methods (see, e.g., Brooks et al. (2011)) are generally regarded as the gold standard approach for sampling from high-dimensional distributions. In particular, MCMC algorithms have been extensively applied within the field of Bayesian statistics to sample from posterior distributions when the posterior density can only be evaluated up to a constant of proportionality. Under mild conditions, it can be shown that asymptotically, the limiting distribution of the samples generated from the MCMC algorithm will converge to the posterior distribution of interest. While theoretically elegant, one of the main drawbacks of MCMC methods is that running the algorithm to stationarity can be prohibitively expensive if the posterior distribution is of a complex form, for example, contains multiple unknown modes. Notable examples of multi-modality include the posterior over model parameters in mixture models (McLachlan and Peel, 2000), deep neural networks (Neal, 2012), and differential equation models (Calderhead and Girolami, 2009).

In this paper, we present the pseudo-extended Markov chain Monte Carlo method as an approach for augmenting the state-space of the original posterior distribution to allow the MCMC sampler to easily move between areas of high posterior density. The pseudo-extended method introduces *pseudo-samples* on the extended space to improve the mixing of the Markov chain. To illustrate how this method works, in Figure 1 we plot a mixture of two univariate Gaussian distributions (*left*). The area of low probability density between the two Gaussians will make it difficult for an MCMC sampler to traverse between them. Using the pseudo-extended approach (as detailed in Section 2), we

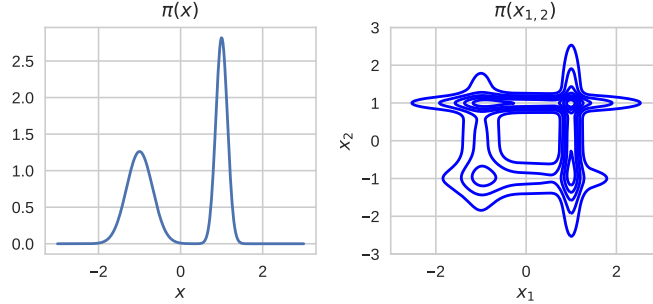

Figure 1: Original target density $\pi(\mathbf{x})$ (left) and extended target (right) with $N = 2$.

can extend the state-space to two dimensions (*right*), where on the extended space, the modes are now connected allowing the MCMC sampler to easily mix between them.

The pseudo-extended framework can be applied for general MCMC sampling, however, in this paper, we focus on using ideas from tempered MCMC (Jasra et al., 2007) to improve multi-modal posterior sampling. Unlike previous approaches which use MCMC to sample from multi-modal posteriors, *i)* we do not require *a priori* information regarding the number, or location, of modes, *ii)* nor do we need to specify a sequence of intermediary tempered distributions (Geyer, 1991).

We show that samples generated using the pseudo-extended method admit the correct posterior of interest as the limiting distribution. Furthermore, once weighted using a *post-hoc* correction step, it is possible to use all pseudo-samples for approximating the posterior distribution. The pseudo-extended method can be applied as an extension to many popular MCMC algorithms, including the random-walk Metropolis (Roberts et al., 1997) and Metropolis-adjusted Langevin algorithm (Roberts and Tweedie, 1996). However, in this paper, we focus on applying the popular Hamiltonian Monte Carlo (HMC) algorithm (Neal, 2010) within the pseudo-extended framework and show that this leads to improved posterior exploration compared to standard HMC.

## 2 The Pseudo-Extended Method

Let $\pi$ be a target probability density on $\mathbb{R}^d$ defined for all $\mathbf{x} \in \mathcal{X} := \mathbb{R}^d$ by

$$\pi(\mathbf{x}) := \frac{\gamma(\mathbf{x})}{Z} = \frac{\exp\{-\phi(\mathbf{x})\}}{Z}, \tag{1}$$

where $\phi : \mathcal{X} \to \mathbb{R}$ is a continuously differentiable function and $Z$ is the normalizing constant. Throughout, we will refer to $\pi(\mathbf{x})$ as the target density. In the Bayesian setting, this would be the posterior, where for data $\mathbf{y} \in \mathcal{Y}$, the likelihood is denoted as $p(\mathbf{y}|\mathbf{x})$ with parameters $\mathbf{x}$ assigned a prior density $\pi_0(\mathbf{x})$. The posterior density of the parameters given the data is derived from Bayes theorem $\pi(\mathbf{x}) = p(\mathbf{y}|\mathbf{x})\pi_0(\mathbf{x})/p(\mathbf{y})$, where the marginal likelihood $p(\mathbf{y})$ is the normalizing constant $Z$, which is typically not available analytically.

We extend the state-space of the original target distribution eq. (1) by introducing $N$ *pseudo-samples*, $\mathbf{x}_{1:N} = \{\mathbf{x}_i\}_{i=1}^N$, where the extended-target distribution $\pi^N(\mathbf{x}_{1:N})$ is defined on $\mathcal{X}^N$. The pseudo-samples act as auxiliary variables, where for each $\mathbf{x}_i$, we introduce an instrumental distribution $q(\mathbf{x}_i) \propto \exp\{-\delta(\mathbf{x}_i)\}$ with support covering that of $\pi(\mathbf{x})$. In a similar vein to the *pseudo-marginal MCMC* algorithm (Beaumont, 2003; Andrieu and Roberts, 2009) our extended-target, including the auxiliary variables, is now of the form,

$$\pi^N(\mathbf{x}_{1:N}) := \frac{1}{N}\sum_{i=1}^N \pi(\mathbf{x}_i)\prod_{j \neq i} q(\mathbf{x}_j) = \frac{1}{Z}\left\{\frac{1}{N}\sum_{i=1}^N \frac{\gamma(\mathbf{x}_i)}{q(\mathbf{x}_i)}\right\} \times \prod_i q(\mathbf{x}_i), \tag{2}$$

where $\gamma(\cdot)$ and $Z$ are defined in eq. (1). In pseudo-marginal MCMC, $q(\cdot)$ is an instrumental distribution used for importance sampling to compute unbiased estimates of the intractable normalizing constant (see Section 2.2 for details). However, with the pseudo-extended method we use $q(\cdot)$ to improve the mixing of the MCMC algorithm. Additionally, unlike pseudo-marginal MCMC, we do not require that $q(\cdot)$ can be sampled from; a fact that we will exploit in Section 3.

In the case where $N = 1$, our extended-target eq. (2) simplifies back to the original target $\pi(\mathbf{x}) = \pi^N(\mathbf{x}_{1:N})$ eq. (1). For $N > 1$, the resulting marginal distribution of the $i$th pseudo-sample is a mixture between the target and the instrumental distribution

$$\pi^N(\mathbf{x}_i) = \frac{1}{N}\pi(\mathbf{x}_i) + \frac{N-1}{N}q(\mathbf{x}_i).$$

We then use a *post-hoc* weighting step to convert the samples from the extended-target to samples from the original target of interest $\pi(\mathbf{x})$. In Theorem 2.1, we show that samples from the extended target give unbiased expectations of arbitrary functions $f$, under the target of interest $\pi$.

**Theorem 2.1.** *Let $\mathbf{x}_{1:N}$ be distributed according to the extended-target $\pi^N$. Weighting each sample with self-normalized weights proportional to $\gamma(\mathbf{x}_i)/q(\mathbf{x}_i)$, for $i = 1, \ldots, N$ gives samples from the target distribution, $\pi(\mathbf{x})$, in the sense that, for an arbitrary integrable $f$,*

$$\mathbb{E}_{\pi^N}\left[\frac{\sum_{i=1}^{N} f(\mathbf{x}_i)\gamma(\mathbf{x}_i)/q(\mathbf{x}_i)}{\sum_{i=1}^{N}\gamma(\mathbf{x}_i)/q(\mathbf{x}_i)}\right] = \mathbb{E}_\pi[f(\mathbf{x})]. \tag{3}$$

The proof follows from the invariance of particle Gibbs (Andrieu et al., 2010) and is given in Section A of the Supplementary Material.

## 2.1 Pseudo-extended Hamiltonian Monte Carlo

We use an MCMC algorithm to sample from the pseudo-extended target eq. (2). In this paper, we use the HMC algorithm because of its impressive mixing times, however, a disadvantage of HMC, and other gradient-based MCMC algorithms is that they tend to be mode-seeking and are more prone to getting trapped in local modes of the target. The pseudo-extended framework creates a target where the modes are connected on the extended space, which reduces the mode-seeking behavior of HMC and allows the sampler to move easily between regions of high density.

Recalling that our parameters are $\mathbf{x} \in \mathcal{X} := \mathbb{R}^d$, we introduce artificial momentum variables $\boldsymbol{\rho} \in \mathbb{R}^d$ that are independent of $\mathbf{x}$. The Hamiltonian $H(\mathbf{x}, \boldsymbol{\rho})$, represents the total energy of the system as the combination of the potential function $\phi(\mathbf{x})$, as defined in eq. (1), and kinetic energy $\frac{1}{2}\boldsymbol{\rho}^\top \mathbf{M}^{-1}\boldsymbol{\rho}$,

$$H(\mathbf{x}, \boldsymbol{\rho}) := \phi(\mathbf{x}) + \frac{1}{2}\boldsymbol{\rho}^\top \mathbf{M}^{-1}\boldsymbol{\rho},$$

where $\mathbf{M}$ is a mass matrix and is often set to the identity matrix. The Hamiltonian now augments our target distribution so that we are sampling $(\mathbf{x}, \boldsymbol{\rho})$ from the joint distribution $\pi(\mathbf{x}, \boldsymbol{\rho}) \propto \exp\{H(\mathbf{x}, \boldsymbol{\rho})\} = \pi(\mathbf{x})\mathcal{N}(\boldsymbol{\rho}|0, \mathbf{M})$, which admits the target as the marginal. In the case of the pseudo-extended target eq. (2), the Hamiltonian is,

$$H^N(\mathbf{x}_{1:N}, \boldsymbol{\rho}) = -\log\left[\sum_{i=1}^{N}\exp\{-\phi(\mathbf{x}_i) + \delta(\mathbf{x}_i)\}\right] + \sum_{i=1}^{N}\delta(\mathbf{x}_i) + \frac{1}{2}\boldsymbol{\rho}^\top \mathbf{M}^{-1}\boldsymbol{\rho}, \tag{4}$$

where now $\boldsymbol{\rho} \in \mathbb{R}^{d \times N}$, and $\delta(\mathbf{x})$ is a potential function of the instrumental distribution that is arbitrary but differentiable, eq. (2).

Aside from a few special cases, we generally cannot simulate from the Hamiltonian system eq. (4) exactly (Neal, 2010). Instead, we discretize time using small step-sizes $\epsilon$ and calculate the state at $\epsilon$, $2\epsilon$, $3\epsilon$, etc. using a numerical integrator. Several numerical integrators are available which preserve the volume and reversibility of the Hamiltonian system (Girolami and Calderhead, 2011), the most popular being the *leapfrog* integrator which takes $L$ steps, each of size $\epsilon$, though the Hamiltonian dynamics (pseudo-code is given in the Supplementary Material). After a fixed number of iterations $T$, the algorithm generates samples $(\mathbf{x}_{1:N}^{(t)}, \boldsymbol{\rho}^{(t)})$, $t = 1, \ldots, T$ approximately distributed according to the joint distribution $\pi(\mathbf{x}_{1:N}, \boldsymbol{\rho})$, where after discarding the momentum variables $\boldsymbol{\rho}$, our MCMC samples will be approximately distributed according to the target $\pi^N(\mathbf{x}_{1:N})$. In this paper, we use the No-U-turn sampler (NUTS) introduced by Hoffman and Gelman (2014) as implemented in the STAN (Carpenter et al., 2017) software package to automatically tune $L$ and $\epsilon$.

## 2.2 Connections to pseudo-marginal MCMC

The pseudo-extended target eq. (2) can be viewed as a special case of the pseudo-marginal target of Andrieu and Roberts (2009). In the pseudo-marginal setting, it is (typically) assumed that the target density is of the form $\pi(\boldsymbol{\theta}) = \int_{\mathcal{X}} \pi(\boldsymbol{\theta}, \mathbf{x}) \mathrm{d}\mathbf{x}$, where $\boldsymbol{\theta}$ is some "top-level" parameter, and where $\mathbf{x}$ are latent variables that cannot be integrated out analytically. Using importance sampling, an unbiased Monte Carlo estimate of the target $\tilde{\pi}(\boldsymbol{\theta})$ is computed using latent variable samples $\mathbf{x}_1, \mathbf{x}_2, \ldots, \mathbf{x}_N$ from an instrumental distribution with density $q(\mathbf{x})$ and then approximating the integral as

$$\tilde{\pi}(\boldsymbol{\theta}) := \frac{1}{N} \sum_{i=1}^{N} \frac{\pi(\boldsymbol{\theta}, \mathbf{x}_i)}{q(\mathbf{x}_i)}, \quad \text{where} \quad \mathbf{x}_i \sim q(\cdot).$$

The pseudo-marginal target is then defined, analogously to the pseudo-extended target eq. (2), as

$$\tilde{\pi}^N(\boldsymbol{\theta}, \mathbf{x}) := \frac{1}{N} \sum_{i=1}^{N} \pi(\boldsymbol{\theta}, \mathbf{x}_i) \prod_{j \neq i} q(\mathbf{x}_j), \tag{5}$$

which admits $\pi(\boldsymbol{\theta})$ as a marginal. In the original pseudo-marginal method, the extended-target is sampled from using MCMC, with an independent proposal for $\mathbf{x}$ (corresponding to importance sampling for these variables) and a standard MCMC proposal (e.g., random-walk) used for $\boldsymbol{\theta}$.

There are two key differences between pseudo-marginal MCMC and pseudo-extended MCMC. Firstly, we do not distinguish between latent variables and parameters, and simply view all unknown variables, or parameters, of interest as being part of $\mathbf{x}$. Secondly, we do not use an importance-sampling-based proposal to sample $\mathbf{x}$, but instead, we propose to simulate directly from the pseudo-extended target eq. (2) using HMC as explained in Section 2.1. An important consequence of this is that we can use instrumental distributions $q(\cdot)$ without needing to sample from them. In Section 3 we exploit this fact to construct the instrumental distribution by tempering.

In summary, the pseudo-marginal framework is a powerful technique for sampling from models with *intractable likelihoods*. The pseudo-extended method, on the other hand, is designed for sampling from *complex target distributions*, where the landscape of the target is difficult for standard MCMC samplers to traverse without an exhaustive number of MCMC iterations. In particular, where the target distribution is multi-modal, we show that extending the state-space allows our MCMC sampler to more easily explore the modes of the target.

## 3 Tempering targets with instrumental distributions

In the case of importance sampling, we would choose an instrumental distribution $q(\cdot)$ which closely approximates the target $\pi(\cdot)$. However, this would assume that we could find a tractable instrumental distribution for $q(\cdot)$ which *i)* sufficiently covers the support of the target and *ii)* captures its multi-modality. Approximations, such as the Laplace approximation (Rue et al., 2009) and variational methods (e.g., Bishop (2006), Chapter 10) could be used to choose $q(\cdot)$, however, such approximations tend to be unimodal and not appropriate for approximating a multi-modal target.

A significant advantage of the pseudo-extended framework eq. (2) is that it permits a wide range of potential instrumental distributions. Unlike standard importance sampling, we also do not require $q(\cdot)$ to be a distribution that we can sample from, only that it can be evaluated point-wise up to proportionality. This is a simpler condition to satisfy and allows us to find better instrumental distributions for connecting the modes of the target. In this paper, we utilize a simple approach for choosing the instrumental distribution which does not require a closed-form approximation of the target. Specifically, we create an instrumental distribution by tempering the target.

Tempering has previously been utilized in the MCMC literature to improve the sampling of multi-modal targets. Here we use a technique inspired by Graham and Storkey (2017) (see Section 3), where we consider the family of approximating distributions,

$$\Pi := \left\{ \pi_\beta(\mathbf{x}) = \frac{\gamma_\beta(\mathbf{x})}{Z(\beta)} : \beta \in (0, 1] \right\}, \tag{6}$$

where $\gamma_\beta(\mathbf{x}) = \exp\{-\beta \phi(\mathbf{x})\}$ can be evaluated point-wise and $Z(\beta)$ is typically intractable.

We will construct an extended target distribution $\pi^N(\mathbf{x}_{1:N}, \beta_{1:N})$ on $\mathcal{X}^N \times (0,1]^N$ with $N$ pairs $(\mathbf{x}_i, \beta_i)$, for $i = 1, \ldots, N$. This target distribution will be constructed in such a way that the marginal distribution of each $\mathbf{x}_i$ is a mixture, with components selected from $\Pi$. This will typically make the marginal distribution more diffuse than the target $\pi$ itself, encouraging better mixing.

If we let $q(\mathbf{x}, \beta) = \pi_\beta(\mathbf{x})q(\beta)$ and choose $q(\beta) = \frac{Z(\beta)g(\beta)}{C}$, where $g(\beta)$ can be evaluated point-wise and $C$ is a normalizing constant, then we can cancel the intractable normalizing constants $Z(\beta)$,

$$q(\mathbf{x}, \beta) = \frac{\gamma_\beta(\mathbf{x})g(\beta)}{C}. \tag{7}$$

The joint instrumental $q(\mathbf{x}, \beta)$ does not admit a closed-form expression and in general we cannot sample from it. However, we do not need to sample from it, as we instead use an MCMC algorithm on the extended-target which only requires that $q(\mathbf{x}, \beta)$ can be evaluated point-wise, up to a constant of proportionality. Under the instrumental proposal eq. (7), the pseudo-extended target eq. (2) is now

$$\pi^N(\mathbf{x}_{1:N}, \beta_{1:N}) := \frac{1}{N} \sum_{i=1}^{N} \pi(\mathbf{x}_i)\pi(\beta_i) \prod_{j \neq i} q(\mathbf{x}_j, \beta_j) \tag{8}$$

$$= \frac{1}{ZC^{N-1}} \left\{ \frac{1}{N} \sum_{i=1}^{N} \frac{\gamma(\mathbf{x}_i)\pi(\beta_i)}{\gamma_{\beta_i}(\mathbf{x}_i)g(\beta_i)} \right\} \prod_{j=1}^{N} \gamma_{\beta_j}(\mathbf{x}_j)g(\beta_j),$$

where $\pi(\beta)$ is some arbitrary user-chosen target distribution for $\beta$. Through our choice of $q(\mathbf{x}, \beta)$, the normalizing constants for the target and instrumental distributions, $Z$ and $C$ respectively are not dependent on $\mathbf{x}$ or $\beta$ and so cancel in the Metropolis-Hastings ratio.

**Related work on tempered MCMC**

Tempered MCMC is the most popular approach to sampling from multi-modal target distributions (see Jasra et al. (2007) for a full review). The main idea behind tempered MCMC is to sample from a sequence of tempered targets,

$$\pi_k(\mathbf{x}) \propto \exp\left\{-\beta_k\phi(\mathbf{x})\right\}, \qquad k = 1, \ldots, K,$$

where $\beta_k$ is a tuning parameter referred to as the *temperature* that is associated with $\pi_k(\mathbf{x})$. A sequence of temperatures, commonly known as the *ladder*, is chosen *a priori*, where $0 = \beta_1 < \beta_2 < \ldots < \beta_K = 1$. The intuition behind tempered MCMC is that when $\beta_k$ is small, the modes of the target are flattened out making it easier for the MCMC sampler to traverse through the regions of low density separating the modes. One of the most popular tempering algorithms is parallel tempering (PT) (Geyer, 1991), where in parallel, $K$ separate MCMC algorithms are run with each sampling from one of the tempered targets $\pi_k(\mathbf{x})$. Samples from neighboring Markov chains are exchanged (i.e. sample from chain $k$ exchanged with chain $k-1$ or $k+1$) using a Metropolis-Hastings step. These exchanges improve the convergence of the Markov chain to the target of interest $\pi(\mathbf{x})$, however, information from low $\beta_k$ targets is often slow to traverse up the temperature ladder. There is also a serial version of this algorithm, known as simulated tempering (ST) (Marinari and Parisi, 1992). An alternative approach is annealed importance sampling (AIS) (Neal, 2001), which draws samples from a simple base distribution and then, via a sequence of intermediate transition densities, moves the samples along the temperature ladder giving a weighted sample from the target distribution. Generally speaking, these tempered approaches can be very difficult to apply in practice often requiring extensive tuning. In the case of PT, the user needs to choose the number of parallel chains $K$, temperature schedule, step-size for each chain and the number of exchanges at each iteration.

Our proposed tempering scheme is closely related to the continuously-tempered HMC algorithm of Graham and Storkey (2017). They propose to run HMC on a distribution similar to eq. (7) and then apply an importance weighting as a post-correction to account for the different temperatures. It thus has some resemblance with ST, in the sense that a single chain is used to explore the state space for different temperature levels. On the contrary, for our proposed pseudo-extended method, the distribution eq. (7) is not used as a target, but merely as an instrumental distribution to construct the pseudo-extended target eq. (8). The resulting method, therefore, has some resemblance with PT, since we propagate $N$ pseudo-samples in parallel, all possibly exploring different temperature levels. Furthermore, by mixing in part of the actual target $\pi$ we ensure that the samples do not simultaneously "drift away" from regions with high probability under $\pi$.

Graham and Storkey (2017) propose to use a variational approximation to the target, both when defining the family of distributions eq. (6) and for choosing the function $g(\beta)$. This is also possible with the pseudo-extended method, but we do not consider this possibility here for brevity. Finally, we note that in the pseudo-extended method the temperature parameter $\beta$ can be estimated as part of the MCMC scheme, rather than pre-tuning it as a sequence of fixed temperatures. This is advantageous because using a coarse grid of temperatures can cause the sampler to miss modes of the target, whereas a fine grid of temperatures leads to a significantly increased computational cost of running the sampler.

## 4 Experiments

We compare the pseudo-extended method on three test models. The first two (Sections 4.1 and 4.2) are chosen to show how the pseudo-extended method performs on simulated data when the target is multi-modal. The third example (Section 4.3) is a sparsity-inducing logistic regression model, where multi-modality occurs in the posterior from three real-world datasets. We compare against popular competing algorithms from the literature, including methods discussed in Section 3.

All simulations for the pseudo-extended method use the tempered instrumental distribution and thus the pseudo-extended target is given by eq. (8). For each simulation study, we set $\pi(\beta) \propto 1$, $g(\beta) \propto 1$ and use a logit transformation for $\beta$ to map the parameters onto the unconstrained space. Additionally, we consider the special case of pseudo-extended HMC where $\beta$ is fixed along a temperature ladder (akin to parallel tempering). The pseudo-extended HMC method is implemented within STAN [1]

### 4.1 Mixture of Gaussians

**Background:** We consider a popular example from the literature (Kou et al., 2006; Tak et al., 2016), where the target is a mixture of 20 bivariate Gaussians,

$$\pi(\mathbf{x}) = \sum_{j=1}^{20} \frac{w_j}{2\pi\sigma_j^2} \exp\left\{ \frac{-1}{2\sigma_j^2}(\mathbf{x} - \boldsymbol{\mu}_j)^\top (\mathbf{x} - \boldsymbol{\mu}_j) \right\},$$

and where $\{\boldsymbol{\mu}_1, \boldsymbol{\mu}_2, \ldots, \boldsymbol{\mu}_{20}\}$ are specified in Kou et al. (2006). We compare the pseudo-extended sampler against parallel tempering (PT) (Geyer, 1991), repelling-attracting Metropolis (RAM) (Tak et al., 2016) and the equi-energy (EE) MCMC sampler (Kou et al., 2006), all of which are designed for sampling from multi-modal distributions.

**Setup:** We consider two simulation settings. In Scenario (a) each mixture component has weight $w_j = 1/20$ and variance $\sigma_j^2 = 1/100$ resulting in well-separated modes with most modes more than 15 standard deviations apart. In Scenario (b) the weights $w_j = 1/||\boldsymbol{\mu}_j - (5,5)^\top||$ and variances $\sigma_j^2 = ||\boldsymbol{\mu}_j - (5,5)^\top||/20$ are unequal where the modes far from (5,5) have a lower weight with larger variance, creating regions of higher density between distant modes (see Figure 2 with further discussion in the Supplementary Material).

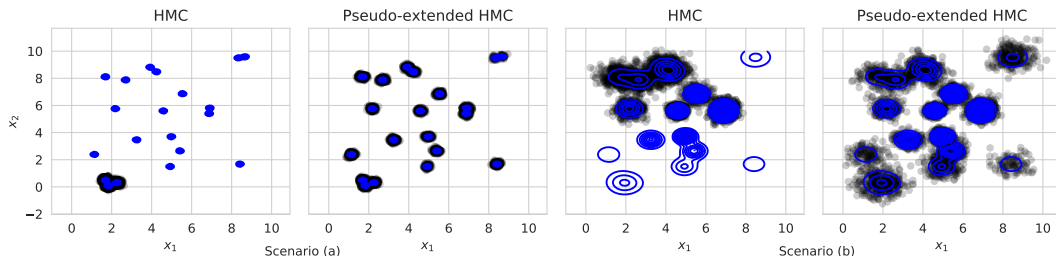

Figure 2: 10,000 samples drawn from the the target under scenario (a) (left) and scenario (b) (right) using the HMC and pseudo-extended HMC samplers.

**Results:** Table 1 gives the root mean squared error (RMSE) of the Monte Carlo estimates, over 20 independent simulations, for the first and second moments. Each sampler was run for 50,000

iterations (after burn-in) and the specific tuning details for the temperature ladder of PT and the energy rings for EE are given in Kou et al. (2006). All the samplers perform worse under Scenario (a) where the modes are well-separated, the HMC sampler is only able to explore the modes locally clustered together, whereas the pseudo-extended HMC sampler is able to explore all of the modes with the same number of iterations (see Section C of the Supplementary Material for posterior plots). Under Scenario (b), there is a higher density region separating the modes making it easier for the HMC sampler to move between the mixture components. While not reported here, the HMC samplers produce Markov chains with significantly reduced auto-correlation compared to the EE and RAM samplers, which both rely on random-walk updates. We note from Table 1 that increasing the number of pseudo-samples leads to improved estimates, but at an increased computational cost. In the Supplementary Material we show that when taking account for computational cost, the optimal number of pseudo-samples is $2 \leq N \leq 5$. Additionally, we can fix rather than estimate $\beta$ and Table 2 in the Supplementary Material shows that this can lead to a small improvement in RMSE if $\beta$ is correctly tuned, but can also (and often does) lead to poorer RMSE if $\beta$ is not well tuned. The conclusion therefore is that it is better to jointly estimate $\pi^N(\mathbf{x}_{1:N}, \beta_{1:N})$ in the absence of *a priori* knowledge of an optimal $\beta$.

Table 1: Root mean-squared error of moment estimates for two mixture scenarios. Results are calculated over 20 independent simulations and reported to two decimal places.

| | Scenario (a) | | | | Scenario (b) | | | |
|---|---|---|---|---|---|---|---|---|
| | $\mathbb{E}[\mathbf{X}_1]$ | $\mathbb{E}[\mathbf{X}_2]$ | $\mathbb{E}[\mathbf{X}_1^2]$ | $\mathbb{E}[\mathbf{X}_2^2]$ | $\mathbb{E}[\mathbf{X}_1]$ | $\mathbb{E}[\mathbf{X}_2]$ | $\mathbb{E}[\mathbf{X}_1^2]$ | $\mathbb{E}[\mathbf{X}_2^2]$ |
| RAM | 0.09 | 0.10 | 0.90 | 1.30 | 0.04 | 0.04 | 0.26 | 0.34 |
| EE | 0.11 | 0.14 | 1.14 | 1.48 | 0.07 | 0.09 | 0.75 | 0.84 |
| PT | 0.18 | 0.28 | 1.82 | 2.89 | 0.12 | 0.13 | 1.15 | 1.22 |
| HMC | 2.69 | 3.96 | 24.69 | 33.65 | 0.27 | 0.51 | 3.12 | 4.80 |
| PE (N=2) | 0.11 | 0.10 | 1.11 | 1.01 | 0.05 | 0.08 | 0.46 | 0.86 |
| PE (N=5) | 0.04 | 0.05 | 0.37 | 0.45 | 0.04 | 0.02 | 0.18 | 0.36 |
| PE (N=10) | 0.03 | 0.03 | 0.28 | 0.23 | **0.02** | 0.02 | **0.10** | 0.32 |
| PE (N=20) | **0.02** | **0.02** | **0.15** | **0.21** | 0.03 | **0.01** | 0.15 | **0.23** |

### 4.2 Boltzmann machine relaxations

**Background:** Sampling from a Boltzmann machine distribution (Jordan et al., 1999) is a challenging inference problem from the statistical physics literature. The probability mass function,

$$P(\mathbf{s}) = \frac{1}{Z_b} \exp\left\{\frac{1}{2}\mathbf{s}^\top \mathbf{W}\mathbf{s} + \mathbf{s}^\top \mathbf{b}\right\}, \quad \text{with} \quad Z_b = \sum_{\mathbf{s} \in \mathcal{S}} \exp\left\{\frac{1}{2}\mathbf{s}^\top \mathbf{W}\mathbf{s} + \mathbf{s}^\top \mathbf{b}\right\}, \quad (9)$$

is defined on the binary space $\mathbf{s} \in \{-1, 1\}^{d_b} := \mathcal{S}$, where $\mathbf{W}$ is a $d_b \times d_b$ real symmetric matrix and $\mathbf{b} \in \mathbb{R}^{d_b}$ are the model parameters. Sampling from this distribution typically requires Gibbs steps (Geman and Geman, 1984) which tend to mix very poorly as the states can be strongly correlated when the Boltzmann machine has high levels of connectivity (Salakhutdinov, 2010). HMC methods have been shown to perform significantly better than Gibbs sampling when the states of the target distribution are highly correlated (Girolami and Calderhead, 2011). Unfortunately, HMC is generally restricted to sampling on continuous spaces. Using the *Gaussian integral trick* (Hertz et al., 1991), we introduce auxiliary variables $\mathbf{x} \in \mathbb{R}^d$ and transform the problem to sampling from $\pi(\mathbf{x})$ rather than eq. (9) (see Section D in the Supplementary Material for full details).

**Setup:** We let $\mathbf{b} \sim \mathcal{N}(0, 0.1^2)$ and set $\mathbf{W} = \mathbf{R}\text{diag}(\mathbf{e})\mathbf{R}^\top$, with diagonal elements set to zero, and simulate a $d_b \times d_b$ random orthogonal matrix for $\mathbf{R}$ (Stewart, 1980). $\mathbf{e}$ is a vector of eigenvalues, with $e_i = \lambda_1 \tanh(\lambda_2 \eta_i)$ and $\eta_i \sim \mathcal{N}(0, 1)$, for $i = 1, 2, \ldots, d_b$. We set $d_b = 28$ ($d = 27$) and let $(\lambda_1, \lambda_2) = (6, 2)$, as these settings have been shown to produce highly multi-modal distributions (see Figure 3 for an example). We compare the HMC and pseudo-extended (PE) HMC algorithms against annealed importance sampling (AIS), simulated tempering (ST), and the continuously-tempered HMC algorithm of Graham and Storkey (2017) (GS). Full set-up details are given in the Supplementary Material.

**Results:** We can analytically derive the first two moments of the Boltzmann distribution (see Section D of the Supplementary Material for details), and in Figure 4 we give the RMSE of the moment

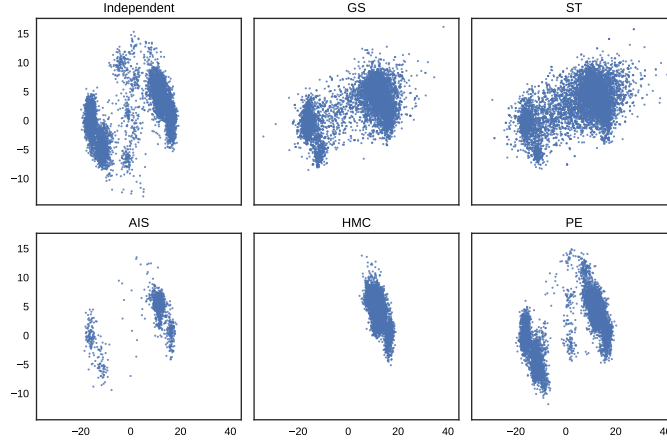

Figure 3: Two-dimensional projection of $10,000$ samples drawn from the target using each of the proposed methods, where the first plot gives the ground-truth sampled directly from the Boltzmann machine relaxation distribution. A temperature ladder of length 1,000 was used for both simulated tempering and annealed importance sampling.

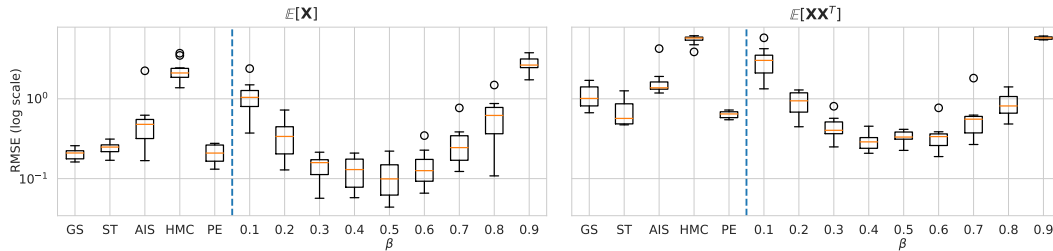

Figure 4: Root mean squared error (log scale) of the first and second moment of the target taken over 10 independent simulations and calculated for each of the proposed methods. Results labeled [0.1-0.9] correspond to pseudo-extended MCMC with fixed $\beta = [0.1 - 0.9]$.

approximations taken over 10 independent runs. These results support the conclusion that better exploration of the target space leads to improved estimation of integrals of interest. Additionally, we note that fixing $\beta$ can produce lower RMSE for PE as we reduce the number of parameters that need to be estimated. However, fixing $\beta$ poorly (e.g. $\beta = 0.1$ in this case) can lead to an increase in RMSE, whereas estimating $\beta$ as part of the inference procedure gives a balanced RMSE result. Further simulations are given in the Supplementary Material which includes plots of posterior samples and the effect of varying the number of pseudo-samples. When taking into account the computational cost, the RMSE is minimized when $2 \leq N \leq 5$, which corroborates with the conclusion from the mixture of Gaussians example (Section 4.1).

### 4.3 Sparse logistic regression with horseshoe priors

**Background:** We apply the pseudo-extended approach to the problem of sparse Bayesian inference. This is a common problem in statistics and machine learning, where the number of parameters to be estimated is much larger than the data used to fit the model. Taking a Bayesian approach, we can use shrinkage priors to shrink model parameters to zero and prevent the model from over-fitting to the data. There are a range of shrinkage priors presented in the literature (Griffin and Brown, 2013) and here we use the horseshoe prior (Carvalho et al., 2010), in particular, the regularized horseshoe as proposed by Piironen and Vehtari (2017). From a sampling perspective, sparse Bayesian inference can be challenging as the posterior distributions are naturally multi-modal, where there is a spike at zero (indicating that variable is inactive) and some posterior mass centered away from zero.

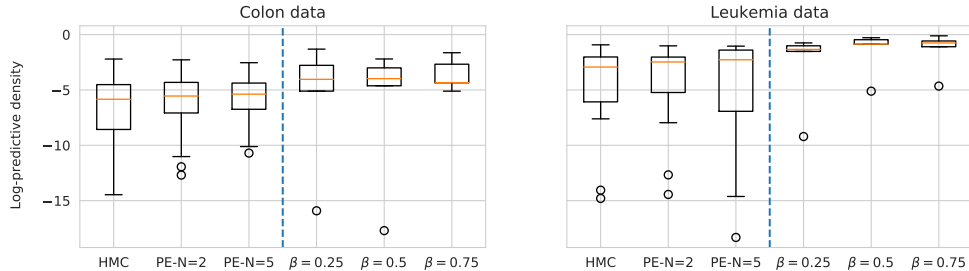

Figure 5: Log-predictive densities on held-out test data (random $20\%$ of full data) for two cancer datasets comparing the HMC and pseudo-extended HMC samplers, with $N = 2$ and $N = 5$. For the case of fixed $\beta = [0.25, 0.5, 0.75]$, the number of pseudo-samples $N = 2$.

**Setup and results:** Following Piironen and Vehtari (2017), we apply the regularized horseshoe prior on a logistic regression model (see Section E of the Supplementary Material for full details). We apply this model to three real-world data sets using micro-array data for cancer classification (prostate data results are given in Section E of the Supplementary Material, see Piironen and Vehtari (2017) for further details regarding the data). We compare the pseudo-extended HMC algorithm against standard HMC and give the log-predictive density on a held-out test dataset in Figure 5. In order to ensure a fair comparison between HMC and pseudo-extended HMC, we run HMC for 10,000 iterations and reduce the number of iterations of the pseudo-extended algorithms (with $N = 2$ and $N = 5$) to give equal total computational cost. The results show that there is an improvement in using the pseudo-extended method, but with a strong performance from standard HMC, which is not surprising in this setting as the posterior density plots (given in the Supplementary Material) show that the posterior modes are close together. As seen in Scenario (b) of Section 4.1, the HMC sampler can usually locate and traverse between modes that are close together. The RMSE for the pseudo-extended method can be improved using a fixed $\beta$, but as noted in the previous examples, $\beta$ is not known *a priori* and fixing it incorrectly can lead to poorer results.

## 5   Discussion

We have introduced the pseudo-extended method as a simple approach for augmenting the target distribution for MCMC sampling. We have shown that the pseudo-extended method can be applied within any general MCMC framework to sample from multi-modal distributions, a challenging scenario for standard MCMC algorithms, and does not require prior knowledge of where, or how many, modes there are in the target. We have shown that a natural instrumental distribution for $q(\cdot)$ is a tempered version of the target, which has the added benefit of automating the choice of instrumental distribution. Alternative instrumental distributions, and methods for estimating the temperature parameter $\beta$, are worthy of further investigation. For example, mixture proposals $q_i$ where each pseudo-variable is associated with a different proposal. Alternatively, the proposal could be *stratified* to encourage each of the pseudo-samples for the temperature parameters $\beta_{1:N}$ to explore different regions of the parameter space. This could be achieved through the choice of the function $g(\cdot)$ (7). If we let $g(\beta_{1:N})$ be a Gaussian distribution, then a valid $N \times N$ covariance matrix $\Sigma$ could be chosen by letting $\Sigma_{ii} = 1$ and $\Sigma_{ij} = -(N-1)^{-1}$, which would induce negative correlation between the pseudo-samples and force the temperatures to be roughly evenly spaced.

## Acknowledgements

CN gratefully acknowledges the support of the UK Engineering and Physical Sciences Research Council grants EP/S00159X/1 and EP/R01860X/1. FL is financially supported by the Swedish Research Council (project 2016-04278), by the Swedish Foundation for Strategic Research (project ICA16-0015) and by the Wallenberg AI, Autonomous Systems and Software Program (WASP) funded by the Knut and Alice Wallenberg Foundation. MF gratefully acknowledges support from the AXA Research Fund and the Agence Nationale de la Recherche (grant ANR-18-CE46-0002).

## Footnotes

[1]https://github.com/chris-nemeth/pseudo-extended-mcmc-code

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
