[Supplementary Material]

# Supplementary Material: Pseudo-Extended Markov Chain Monte Carlo

## A    Proof of Theorem 2.1

We start by assuming that $\mathbf{x}_{1:N}$ are distributed according to the extended-target $\pi^N(\mathbf{x}_{1:N})$. Assuming there exists a measurable function, $f$, we define the expectation of the function over the extended-target as $\mathbb{E}_{\pi^N}\left[\frac{\sum_{i=1}^{N} f(\mathbf{x}_i)\gamma(\mathbf{x}_i)/q(\mathbf{x}_i)}{\sum_{i=1}^{N}\gamma(\mathbf{x}_i)/q(\mathbf{x}_i)}\right]$, where $\gamma(\mathbf{x})$ is the unnormalized target density eq. (1) and $q(\mathbf{x})$ is the instrumental distribution (discussed in Section 2). Using the density for the pseudo-extended target eq. (2), it follows that

$$
\begin{aligned}
\mathbb{E}_{\pi^N}\left[\frac{\sum_{i=1}^{N} f(\mathbf{x}_i)\gamma(\mathbf{x}_i)/q(\mathbf{x}_i)}{\sum_{i=1}^{N}\gamma(\mathbf{x}_i)/q(\mathbf{x}_i)}\right] &= \int \frac{\sum_{i=1}^{N} f(\mathbf{x}_i)\gamma(\mathbf{x}_i)/q(\mathbf{x}_i)}{\sum_{i=1}^{N}\gamma(\mathbf{x}_i)/q(\mathbf{x}_i)}\pi^N(\mathbf{x}_{1:N})\mathrm{d}\mathbf{x}_{1:N} \\
&= \int \frac{\sum_{i=1}^{N} f(\mathbf{x}_i)\gamma(\mathbf{x}_i)/q(\mathbf{x}_i)}{\sum_{i=1}^{N}\gamma(\mathbf{x}_i)/q(\mathbf{x}_i)}\frac{1}{Z}\left\{\frac{1}{N}\sum_{i=1}^{N}\frac{\gamma(\mathbf{x}_i)}{q(\mathbf{x}_i)}\right\}\prod_i q(\mathbf{x}_i)\mathrm{d}\mathbf{x}_{1:N} \\
&= \frac{1}{ZN}\int\left\{\sum_{i=1}^{N} f(\mathbf{x}_i)\frac{\gamma(\mathbf{x}_i)}{q(\mathbf{x}_i)}\right\}\times\prod_i q(\mathbf{x}_i)\mathrm{d}\mathbf{x}_{1:N} \\
&= \frac{1}{N}\int\sum_{i=1}^{N} f(\mathbf{x}_i)\pi(\mathbf{x}_i)\prod_{j\neq i} q(\mathbf{x}_j)\mathrm{d}\mathbf{x}_{1:N} \\
&= \frac{1}{N}\sum_{i=1}^{N}\int f(\mathbf{x}_i)\pi(\mathbf{x}_i)\mathrm{d}\mathbf{x}_i\prod_{j\neq i} q(\mathbf{x}_j) = \mathbb{E}_{\pi}[f(\mathbf{x})]\quad\square
\end{aligned}
$$

## B    Pseudo-extended Hamiltonian Monte Carlo algorithm

---
**Algorithm 1** Pseudo-extended HMC

---
   **Input:** Initial parameters $\mathbf{x}_{1:N}^{(0)}$, step-size $\epsilon$ and trajectory length $L$.
   **for** $t = 1$ **to** $T$ **do**
      Set $\mathbf{y}^{t-1} \leftarrow \mathbf{x}_{1:N}^{t-1}$ {for notational convenience}
      Sample momentum $\boldsymbol{\rho} \sim \mathcal{N}(0, \mathbf{M})$
      Set $\mathbf{y}_1 \leftarrow \mathbf{y}^{t-1}$ and $\boldsymbol{\rho}_1 \leftarrow \boldsymbol{\rho}$
      **for** $l = 1$ **to** $L$ **do**
         $\boldsymbol{\rho}_{l+\frac{1}{2}} \leftarrow \boldsymbol{\rho}_l + \frac{\epsilon}{2}\nabla\log\pi^N(\mathbf{y}_l)$
         $\mathbf{y}_{l+1} \leftarrow \mathbf{y}_l + \epsilon\mathbf{M}^{-1}\boldsymbol{\rho}_{l+\frac{1}{2}}$
         $\boldsymbol{\rho}_{l+1} \leftarrow \boldsymbol{\rho}_{l+\frac{1}{2}} + \frac{\epsilon}{2}\nabla\log\pi^N(\mathbf{y}_{l+1})$
      **end for**
      With probability,

$$\min\left\{1, \exp[H^N(\mathbf{y}^{t-1}, \boldsymbol{\rho}^{t-1}) - H^N(\mathbf{y}_{L+1}, \boldsymbol{\rho}_{L+1})]\right\}$$

      set $\mathbf{x}_{1:N}^t \leftarrow \mathbf{y}_{L+1}$
   **end for**
   **Output:** Samples $\{\mathbf{x}_{1:N}^t\}_{t=1}^T$ from $\pi^N(\mathbf{x}_{1:N})$ and $\mathbb{E}_\pi[f(\mathbf{x})]$ is calculated using eq. (3).

---

### B.1    One-dimensional illustration

Consider a bi-modal target of the form (see Figure 1 (*left*)),

$$\pi(\mathbf{x}) \propto \mathcal{N}(-1, 0.1) + \mathcal{N}(1, 0.02).$$

If there are $N = 2$ pseudo-samples, the pseudo-extended target eq. (2) simplifies to

$$\pi(\mathbf{x}_{1:2}) \propto \gamma(\mathbf{x}_1)q(\mathbf{x}_2) + \gamma(\mathbf{x}_2)q(\mathbf{x}_1),$$

and for the sake of illustration, we choose $q(\mathbf{x}) = \mathcal{N}(0, 2)$.

Density plots for the original and pseudo-extended target are given in Figure 1. On the original target, the modes are separated by a region of low density and an MCMC sampler will therefore only pass between the modes with low probability, thus potentially requiring an exhaustive number of iterations. On the pseudo-extended target, the modes of the original target $\pi(\mathbf{x})$ are now connected on the extended space $\pi(\mathbf{x}_{1,2})$. The instrumental distribution $q$ has the effect of increasing the density in the low probability regions of the target which separate the modes. A higher density between the modes means that the MCMC sampler can now traverse between the modes with higher probability than under the original target.

Figure 6: 10,000 samples from the target (left) and extended target (right) using HMC sampler

In Figure B.1, density plots of the original target are overlayed with samples drawn from the original and pseudo-extended targets using the HMC algorithm, respectively. After 10,000 iterations of the HMC sampler on the original target only one mode is discovered. Applying the same HMC algorithm on the pseudo-extended target, and then weighting the samples (as discussed in Section 2), both modes of the original target are discovered and the samples produce a good empirical approximation to the target.

## C  Mixture of Gaussians

The pseudo-extended sampler with tempered instrumental distributions (Section 3) performs well in both scenarios, where the modes are close or far apart. For the smallest number of pseudo-samples ($N = 2$), the pseudo-extended HMC sampler performs equally as well as the competing methods. Increasing the number of pseudo-samples leads to a decrease in the standard deviation of the moment estimates. However, increasing the number of pseudo-samples also increases the overall computational cost of the pseudo-extended sampler. Figure 7 measures the cost of the pseudo-extended sampler as the average mean squared error (over 20 runs) multiplied by the computational time. From the figure we see that by minimizing the error relative to computational cost, the optimal number of pseudo-samples, under both scenarios, is between 2 and 5. We also note that Figure 7 suggests that the number of pseudo-samples may be problem specific. In Scenario (a), where the modes are well-separated, increasing the number of pseudo-samples beyond 5 does not significantly increase the cost of the sampler, whereas under Scenario (b), using more than 5 pseudo-samples (where the mixture components are easier to explore) introduces a significant increase in the computational cost without a proportional reduction in the error.

We ran the HMC and pseudo-extended HMC ($N = 2$) samplers under the same conditions as in Kou et al. (2006) and Tak et al. (2016), for 10,000 iterations. Figure 2 shows the samples drawn using standard HMC and pseudo-extended HMC. In Scenario (a), where the modes are well-separated, the HMC sampler is only able to explore the modes locally clustered together, whereas the pseudo-extended HMC sampler is able to explore all of the modes, for the same number of iterations. Under Scenario (b), the weights and variances of the mixture components are larger than under Scenario (a), as a result, there is a higher density region separating the modes making it easier for the HMC sampler to move between the mixture components. Compared to the pseudo-extended HMC sampler, the HMC sampler is still not able to explore all of the modes of the target.

Figure 7: Average mean squared error (MSE) (given on the log scale) of the first and second moments taken over 20 independent simulations for varying number of pseudo-samples $N$, where MSE is scaled by computational time (CT) and plotted as MSE $\times$ CT.

The results of Table 1 show that all of the samplers, with the exception of HMC, provide accurate estimates of the first two moments of the target. Under Scenario (a), the HMC sampler produces significantly biased estimates as a result of not exploring all of the modes of the target (see Figure 2), whereas under Scenario (b), while still performing worse than the other samplers, the HMC estimates are significantly less biased as the sampler is able to explore the majority of modes of the target. The RAM and EE samplers perform equally well with PT showing the highest standard deviation of the moment estimates under both scenarios. Under some of the simulations, PT did not explore all of the modes, and as discussed in Kou et al. (2006), parallel tempering has to be carefully tuned to avoid becoming trapped in local modes.

A special case of the pseudo-extended framework is to fix rather than estimate $\beta$. This has the advantage that there are now fewer parameters to estimate, resulting in less Monte Carlo variation. Table 2 provides extended RMSE results, similar to those from Table 1, where $\beta \in \{0.1, 0.2, 0.3, 0.4, 0.5, 0.6, 0.7, 0.8, 0.9\}$. These results show that there is the potential for an improved pseudo-extended sampler (in terms of RMSE), if $\beta$ is well-tuned *a priori*. However, without prior knowledge about the target distribution, it is unlikely that $\beta$ can be appropriately tuned and would therefore require several independent MCMC chains, akin to PT, or an adaptive method to tune $\beta$ during the MCMC sampling.

## D  Boltzmann machine relaxation derivations

The Boltzmann machine distribution is defined on the binary space $\mathbf{s} \in \{-1, 1\}^{d_b} := \mathcal{S}$ with mass function

$$P(\mathbf{s}) = \frac{1}{Z_b} \exp\left\{\frac{1}{2}\mathbf{s}^\top \mathbf{W}\mathbf{s} + \mathbf{s}^\top \mathbf{b}\right\}, \quad Z_b = \sum_{\mathbf{s} \in \mathcal{S}} \exp\left\{\frac{1}{2}\mathbf{s}^\top \mathbf{W}\mathbf{s} + \mathbf{s}^\top \mathbf{b}\right\}, \tag{10}$$

where $\mathbf{b} \in \mathbf{R}^{d_b}$ and $\mathbf{W}$ is a $d_b \times d_b$ real symmetric matrix are the model parameters.

Following the approach of Graham and Storkey (2017) and Zhang et al. (2012), we convert the problem of sampling on the $2^{d_b}$ discrete space to a continuous problem using the Gaussian integral trick (Hertz et al., 1991). We introduce the auxiliary variable $\mathbf{x} \in \mathbb{R}^d$ which follows a conditional Gaussian distribution,

$$\pi(\mathbf{x}|\mathbf{s}) = \frac{1}{(2\pi)^{d/2}} \exp\left\{-\frac{1}{2}(\mathbf{x} - \mathbf{Q}^\top \mathbf{s})^\top (\mathbf{x} - \mathbf{Q}^\top \mathbf{s})\right\}, \tag{11}$$

Table 2: Root mean-squared error of moment estimates for two mixture scenarios. The first row corresponds to the results for pseudo-extended MCMC when $\beta$ is estimated and the remaining cases are for fixed $\beta = [0.1, 0.2, 0.3, 0.4, 0.5, 0.6, 0.7, 0.8, 0.9]$. Results are calculated over 20 independent simulations and reported to two decimal places with bold font indicating the lowest RMSE in each column.

| | | Scenario (a) | | | | Scenario (b) | | | |
|---|---|---|---|---|---|---|---|---|---|
| | | $\mathbb{E}[\mathbf{X}_1]$ | $\mathbb{E}[\mathbf{X}_2]$ | $\mathbb{E}[\mathbf{X}_1^2]$ | $\mathbb{E}[\mathbf{X}_2^2]$ | $\mathbb{E}[\mathbf{X}_1]$ | $\mathbb{E}[\mathbf{X}_2]$ | $\mathbb{E}[\mathbf{X}_1^2]$ | $\mathbb{E}[\mathbf{X}_2^2]$ |
| $\beta$ | N=2 | 0.11 | 0.10 | 1.11 | 1.01 | 0.05 | 0.08 | 0.46 | 0.86 |
| | N=5 | 0.04 | 0.05 | 0.37 | 0.45 | 0.04 | 0.02 | 0.18 | 0.36 |
| | N=10 | 0.03 | 0.03 | 0.28 | 0.23 | 0.02 | 0.02 | **0.10** | 0.32 |
| | N=20 | **0.02** | **0.02** | **0.15** | **0.21** | 0.03 | **0.01** | 0.15 | 0.23 |
| $\beta = 0.1$ | N=2 | 0.91 | 1.31 | 9.96 | 12.43 | 0.03 | 0.04 | 0.34 | 0.40 |
| | N=5 | 0.65 | 0.70 | 7.30 | 7.19 | **0.01** | 0.03 | 0.19 | 0.36 |
| | N=10 | 0.70 | 0.61 | 7.87 | 6.35 | **0.01** | **0.01** | 0.18 | 0.21 |
| | N=20 | 0.69 | 0.49 | 7.84 | 5.68 | **0.01** | **0.01** | 0.19 | **0.15** |
| $\beta = 0.2$ | N=2 | 2.46 | 3.79 | 20.28 | 30.67 | 0.03 | 0.04 | 0.32 | 0.55 |
| | N=5 | 2.71 | 4.08 | 22.20 | 32.33 | **0.01** | 0.03 | 0.25 | 0.29 |
| | N=10 | 2.67 | 4.01 | 21.91 | 31.97 | **0.01** | **0.01** | 0.22 | 0.18 |
| | N=20 | 2.73 | 4.05 | 22.26 | 32.21 | **0.01** | **0.01** | 0.17 | 0.22 |
| $\beta = 0.3$ | N=2 | 2.55 | 4.22 | 21.34 | 32.25 | 0.05 | 0.08 | 0.51 | 0.81 |
| | N=5 | 2.52 | 3.96 | 20.97 | 31.22 | 0.02 | 0.02 | 0.28 | 0.25 |
| | N=10 | 2.64 | 4.09 | 21.74 | 32.37 | **0.01** | 0.03 | 0.13 | 0.32 |
| | N=20 | 2.72 | 4.16 | 22.34 | 32.57 | **0.01** | 0.02 | 0.17 | 0.20 |
| $\beta = 0.4$ | N=2 | 2.59 | 3.71 | 21.03 | 30.99 | 0.05 | 0.11 | 0.55 | 1.16 |
| | N=5 | 2.41 | 3.54 | 19.88 | 29.93 | 0.02 | 0.05 | 0.31 | 0.49 |
| | N=10 | 2.52 | 3.76 | 20.72 | 31.17 | 0.02 | 0.04 | 0.25 | 0.36 |
| | N=20 | 2.73 | 4.13 | 22.37 | 32.51 | 0.02 | 0.02 | 0.18 | 0.22 |
| $\beta = 0.5$ | N=2 | 2.54 | 3.90 | 20.96 | 31.57 | 0.07 | 0.13 | 0.75 | 1.48 |
| | N=5 | 2.38 | 3.93 | 20.03 | 31.41 | 0.03 | 0.07 | 0.39 | 0.76 |
| | N=10 | 2.27 | 3.83 | 19.41 | 30.97 | 0.03 | 0.05 | 0.34 | 0.56 |
| | N=20 | 2.36 | 3.85 | 20.12 | 31.34 | 0.02 | 0.03 | 0.19 | 0.35 |
| $\beta = 0.6$ | N=2 | 2.76 | 4.06 | 23.05 | 31.90 | 0.10 | 0.19 | 1.09 | 1.92 |
| | N=5 | 2.45 | 4.01 | 20.46 | 31.87 | 0.06 | 0.10 | 0.70 | 1.00 |
| | N=10 | 2.35 | 3.77 | 19.63 | 31.00 | 0.05 | 0.07 | 0.63 | 0.73 |
| | N=20 | 2.12 | 3.60 | 18.04 | 30.73 | 0.03 | 0.05 | 0.34 | 0.54 |
| $\beta = 0.7$ | N=2 | 2.50 | 4.12 | 20.85 | 31.98 | 0.15 | 0.25 | 1.75 | 2.76 |
| | N=5 | 2.68 | 4.00 | 21.88 | 32.08 | 0.08 | 0.14 | 0.86 | 1.47 |
| | N=10 | 2.65 | 4.13 | 21.91 | 32.44 | 0.06 | 0.11 | 0.67 | 1.11 |
| | N=20 | 2.67 | 4.01 | 21.97 | 32.10 | 0.04 | 0.07 | 0.52 | 0.81 |
| $\beta = 0.8$ | N=2 | 2.59 | 4.02 | 21.17 | 32.16 | 0.30 | 0.52 | 3.52 | 5.88 |
| | N=5 | 2.71 | 3.97 | 21.94 | 31.75 | 0.10 | 0.16 | 1.25 | 1.91 |
| | N=10 | 2.74 | 4.11 | 22.39 | 32.46 | 0.10 | 0.13 | 1.34 | 1.66 |
| | N=20 | 2.73 | 4.13 | 22.37 | 32.51 | 0.04 | 0.07 | 0.38 | 0.67 |
| $\beta = 0.9$ | N=2 | 2.66 | 4.01 | 21.51 | 31.94 | 0.32 | 0.44 | 3.85 | 5.12 |
| | N=5 | 2.77 | 4.07 | 22.48 | 32.30 | 0.15 | 0.27 | 1.73 | 2.96 |
| | N=10 | 2.73 | 4.13 | 22.38 | 32.49 | 0.14 | 0.19 | 1.67 | 2.28 |
| | N=20 | 2.74 | 4.09 | 22.42 | 32.41 | 0.07 | 0.13 | 0.87 | 1.49 |
| HMC | | 2.69 | 3.96 | 24.69 | 33.65 | 0.27 | 0.51 | 3.12 | 4.80 |

where $\mathbf{Q}$ is a $d_b \times d$ matrix such that $\mathbf{Q}\mathbf{Q}^\top = \mathbf{W} + \mathbf{D}$ and $\mathbf{D}$ is a diagonal matrix chosen to ensure that $\mathbf{W} + \mathbf{D}$ is a positive semi-definite matrix.

Combining eq. (10) and eq. (11) the joint distribution is,

$$\pi(\mathbf{x}, \mathbf{s}) = \frac{1}{(2\pi)^{d/2} Z_b} \exp\left\{ -\frac{1}{2}\mathbf{x}^\top\mathbf{x} + \mathbf{s}^\top\mathbf{Q}\mathbf{x} - \frac{1}{2}\mathbf{s}^\top\mathbf{Q}\mathbf{Q}^\top\mathbf{s} + \frac{1}{2}\mathbf{s}^\top\mathbf{W}\mathbf{s} + \mathbf{s}^\top\mathbf{b} \right\}$$

$$= \frac{1}{(2\pi)^{d/2} Z_b} \exp\left\{ -\frac{1}{2}\mathbf{x}^\top\mathbf{x} + \mathbf{s}^\top(\mathbf{Q}\mathbf{x} + \mathbf{b}) - \frac{1}{2}\mathbf{s}^\top\mathbf{D}\mathbf{s} \right\}$$

$$= \frac{1}{(2\pi)^{d/2} Z_b \exp\left\{\frac{1}{2}\text{Tr}(\mathbf{D})\right\}} \exp\left\{ -\frac{1}{2}\mathbf{x}^\top\mathbf{x} \right\} \prod_{k=1}^{d_b} \exp\left\{ s_k(\mathbf{q}_k^\top\mathbf{x} + b_k) \right\},$$

where $\{\mathbf{q}_k^\top\}_{k=1}^{d_b}$ are the rows of $\mathbf{Q}$. The key feature of this trick is that the $\frac{1}{2}\mathbf{s}^\top\mathbf{W}\mathbf{s}$ term cancel. On the joint space the binary variables $\mathbf{s}$ variables are now decoupled and can be summed over independently to give the marginal density,

$$\pi(\mathbf{x}) = \frac{2^{d_b}}{(2\pi)^{d/2} Z_b \exp\left\{\frac{1}{2}\text{Tr}(\mathbf{D})\right\}} \exp\left\{ -\frac{1}{2}\mathbf{x}^\top\mathbf{x} \right\} \prod_{i=k}^{d_b} \cosh(\mathbf{q}_k^\top\mathbf{x} + b_k),$$

which is referred to as the *Boltzmann machine relaxation* density, which is a Gaussian mixture with $2^{d_b}$ components.

We can rearrange the terms in the Boltzmann machine relaxation density to match our generic target $\pi(\mathbf{x}) = Z^{-1}\exp\{-\phi(\mathbf{x})\}$, eq. (1), where

$$\phi(\mathbf{x}) = \frac{1}{2}\mathbf{x}^\top\mathbf{x} - \sum_{k=1}^{d_b} \log\cosh(\mathbf{q}_k^\top\mathbf{x} + b_k),$$

and the normalizing constant is directly related to the Boltzmann machine distribution

$$\log Z = \log Z_b + \frac{1}{2}\text{Tr}(\mathbf{D}) + \frac{d}{2}\log(2\pi) - d_b\log 2.$$

Converting a discrete problem onto the continuous space does not automatically guarantee that sampling from the continuous space will be any easier than on the discrete space. In fact, if the elements of $\mathbf{D}$ are large, then on the relaxed space, the modes of the $2^{d_b}$ mixture components will be far apart making it difficult for an MCMC sampler to explore the target. Following Zhang et al. (2012), for the experiments in this paper we select $\mathbf{D}$ by minimizing the maximum eigenvalue of $\mathbf{W} + \mathbf{D}$ which has the effect of decreasing the separation of the mixture components on the relaxed space.

Finally, the first two moments of the relaxed distribution can be directly related to their equivalent moments for the Boltzmann machine distribution by

$$\mathbb{E}[\mathbf{X}] = \int_\mathcal{X} \mathbf{x} \sum_{\mathbf{s}\in\mathcal{S}} \pi(\mathbf{s}|\mathbf{x})P(\mathbf{s})\mathrm{d}\mathbf{x} = \sum_{\mathbf{s}\in\mathcal{S}}\left[\int_\mathcal{X} \mathbf{x}\mathcal{N}(\mathbf{x}|\mathbf{Q}^\top\mathbf{s}, \mathbf{I})\mathrm{d}\mathbf{x}P(\mathbf{s})\right] = \mathbb{E}[\mathbf{Q}^\top\mathbf{S}] = \mathbf{Q}^\top\mathbb{E}[\mathbf{S}],$$

$$\mathbb{E}\left[\mathbf{X}\mathbf{X}^\top\right] = \sum_{\mathbf{s}\in\mathcal{S}}\left[\int_\mathcal{X} \mathbf{x}\mathbf{x}^\top\mathcal{N}(\mathbf{x}|\mathbf{Q}^\top\mathbf{s}, \mathbf{I})\mathrm{d}\mathbf{x}P(\mathbf{s})\right] = \mathbb{E}[\mathbf{Q}^\top\mathbf{S}\mathbf{S}^\top\mathbf{Q} + \mathbf{I}] = \mathbf{Q}^\top\mathbb{E}\left[\mathbf{S}\mathbf{S}^\top\right] + \mathbf{I}.$$

For the MCMC simulation comparison given in Section 4.2, we compare our pseudo-extended (PE) method against HMC, annealed importance sampling (AIS), simulated tempering (ST) and the Graham and Storkey (2017) (GS) algorithm. For the setting where $d_B = 28$ we can draw independent samples from the Boltzmann distribution eq. (9), if $d_B$ where any large, then this would not be possible. We run each of the competing algorithms for 10,000 iterations and for PE, we test $N = \{2, 5, 10, 15, 20\}$ (see Figure 8) but in Figure 3 we only plot the results for $N = 5$. For ST and AIS, both of which require a temperature ladder $\beta_t$, we used a ladder of length 1,000 with equally-spaced uniform $[0, 1]$ intervals.

In the simulations, all of the algorithms were hand tuned to achieve optimal performance with a temperature ladder of length 1,000 used for both simulated tempering and annealed importance

sampling. The final 10,000 iterations for each algorithm were used to calculate the root mean squared errors of the estimates of the first two moments, taken over 10 independent runs, and are given in Figure 4. The multi-modality of the target makes it difficult for the standard HMC algorithm to adequately explore the target, and as shown in Figure 3, the HMC algorithm is not able to traverse the modes of the target. The remaining algorithms perform reasonably well in approximating the first two moments of the distribution with some evidence supporting the improved performance of the pseudo-extended algorithm and simulated tempering approach.

As noted in the mixture of Gaussians example (Section 4.1), increasing the number of pseudo-samples improves the accuracy of the pseudo-extended method, but at a computational cost which grows linearly with $N$. When choosing the number of pseudo-samples it is sensible that $N$ increases linearly with the dimension of the target. However, taking into account computational cost (Figure 8), a significantly smaller number of pseudo-samples can be used while still achieving a high level of sampling accuracy.

Figure 8: Average mean squared error (MSE) (given on the log scale) taken over 10 independent simulations with varying number of pseudo-samples $N$, where the MSE is scaled by computational time as MSE $\times$ CT

# E  Sparse logistic regression plots

We consider the following logistic regression model for data $y \in 0, 1$,
$$Pr(Y = y) = p^y(1 - p)^1 - y,$$
where
$$p = \frac{1}{1 + \exp(\mathbf{z}^\top \mathbf{x})}$$
and $\mathbf{z}$ are covariates. In this setting our parameter of interest $\mathbf{x}$ is the model coefficient, and recalling that $\mathbf{x} = (x_1, \dots, x_d)$, we can define a regularized horseshoe prior (Piironen and Vehtari, 2017) on each of the coefficients as,
$$x_j | \lambda_j, \tau, c \sim \mathcal{N}(0, \tau^2 \tilde{\lambda}_j^2), \quad \tilde{\lambda}_j^2 = \frac{c^2 \lambda_j^2}{c^2 + \tau^2 \lambda_j^2},$$
$$\lambda_j \sim C^+(0, 1), j = 1, \dots, d,$$
where $c > 0$ is a constant (for which we follow Piironen and Vehtari (2017) in choosing) and $C^+$ is a half-Cauchy distribution. To give an indication of how this prior behaves, when $\tau^2 \lambda_j^2 << c^2$, the coefficient $x_j$ is close to zero and the regularized horseshoe prior (above) approaches the original horseshoe (Carvalho et al., 2010). Alternatively, when $\tau^2 \lambda_j^2 >> c^2$, the coefficient $x_j$ moves away from zero and the regularizing feature of this prior means that it approaches $\mathcal{N}(0, c^2)$.

Figure 9 gives the posterior density plots for a random subset of the model parameters for each dataset. We can see from these plots that the posteriors are mostly uni-modal with some posterior mass centered at zero. This is a common trait of horseshoe and similar priors for inducing sparsity, where the point-mass at zero indicates that the variable is turned-off (mass at zero), or contains some positive posterior mass elsewhere. We also note that, unlike the examples given in Sections C and 4.2, the posterior modes are close together. For this reason, it is unsurprising that the HMC algorithm is able to accurately explore the posterior space, and as a result, produce accurate log-predictive estimates (as seen in Figure 5). Additionally, see Figure 10 for log-predictive results on the prostate dataset.

(a) Colon

(b) Leukemia

(c) Prostate

Figure 9: Plots of marginal posterior densities for a random subsample of variables. Each column represents a different variable and each row is a different MCMC sampler, HMC, PE-HMC (N=2) and PE-HMC (N=5), respectively

Figure 10: Log-predictive density on held-out test data (random $20\%$ of full data) for the prostate cancer dataset comparing the HMC and pseudo-extended HMC samplers, with $N = 2$ and $N = 5$. For the case of fixed $\beta = [0.25, 0.5, 0.75]$, the number of pseudo-samples $N = 2$.