[Reviews · NeurIPS 2019]

Reviewer 1



Update: I have read the author response and am satisfied with the commitment to elaborate on \beta and \pi and to simplify the Stan PE code with a "pseudo-extended" function. ------- # Overview Sampling from multi-modal distributions is a challenging problem especially in multivariate regimes. This paper presents a new MCMC sampling method called pseudo-extended MCMC that uses an instrumental distribution to projects the data into a higher-dimensional space where the modes are connected, making it easier for the sampler to mix. A default instrumental distribution based on tempering is provided. Technical details are complete. The method is compared to existing baselines showing efficacy on three benchmark datasets. # Clarity and Quality The paper is very well-written. The paper is well-placed within the existing literature. Related methods (pseudo-marginal MCMC and continuously-tempered HMC) are appropriately cited and compared to at a technical and intuitive level, as are several sampling baselines. The experiments are easy to understand and comprehensive. The appendix contains a deferred proof, a description of the algorithm, and further experimental details and plots. Code is given for each experiment showing how the authors used Stan to implement the sampler and some of the baselines (and references to the code for RAM, EE and PT). Overall I believe this is a high-quality submission and I recommend for the paper to be accepted. # Questions 1. The paper focuses on HMC sampling. Do you believe that PE can be used in MH sampling or in SMC sampling? With discrete variables? 2. After Eq(8) it is mentioned that \pi(\beta) is an arbitrary user-chosen target and later than \pi(\beta) and g(\beta) are set equal to 1 for the experiments. Could you please provide intuition on what role \pi(\beta) and g(\beta) play, with respect to characteristics of the inference problem? From Fig. 2, \beta plays a key role. How do you recommend setting \pi and g to best estimate \beta? 3. Could you please explain briefly how the models you wrote in Stan ensure that the sampler implements Algorithm 1 from Appendix B? 4. Looking through the Stan programs, the PE sampling code tends to be longer and more complicated. Have you thought about ways to reduce the complexity of the sampling code so that there is a clearer separation between modeling and inference?

Reviewer 2



The idea of this paper is novel but tightly linked to the popular 'pseudo-marginal MCMC' work (2009). Nonetheless, it has key differences which are explained in the paper and addresses a different sampling problem (i.e. transition in multimodal distributions rather than dealing with intractable likelihoods). As such, I find the idea of this work interesting, simple and new. I also think this paper can potentially have the same impact that the aforementioned work had. The quality of the paper is high and the proposed method and the relevant works are explained clearly.

Reviewer 3



POST AUTHOR RESPONSE: The authors haven't really addressed the main criticism that the experiments 4.1 and 4.2 appear very trivial with no conditioning on data. It is definitely helpful to pick examples where the exact answer is known, but simplifying them to the point that they appear insignificant undercuts the impact of the results. Just my 2 cents! Showing additional plots along the lines of the Stan ADVI paper would be very convincing to the audience, but without actually seeing these I can't increase my rating. Based on the technical descriptions alone this paper is above the threshold of acceptance. However, for this to be a top paper it must prove the point without doubt with experiments. ORIGINAL REVIEW BELOW: Originality: The specific form of tempered distribution that is proposed in the paper is quite original. The paper proposes to draw samples from a mixture of distributions each of which is a product of the target distribution and tempered versions of the target with priors on the temperatures as well (Section 3). This unique ensemble distribution preserves the differentiability property of the original distribution thus allowing standard methods such as HMC to draw samples from the distribution, and at the same tine allows exploration of the posterior space through low density regions. The aspect where the temperatures themselves are treated as random variables make this a very practical idea. Quality: The paper is technically sound, but it is not very well supported by the experiments. The experiments in 4.1 and 4.2 use the RMSE error of the target variables which is quite unusual. The more appropriate accuracy measure is the predictive likelihood of held out data vs inference time. Further, there is no conditioning on data in 4.1 and 4.2 so these are very trivial experiments in that all we are trying to do is generate a sample from the prior. In 4.3, it is not clear that the posterior modes are significantly separated and the log likelihoods in Figure 3 doesn't show any significant improvement either. Also, no timing information is provided. Clarity: The ideas in the paper are clearly described. Significance: The tempering solution is quite unique, but it is limited to producing weighted samples. The idea of using a product of distributions and temperatures would certainly be picked up by other researchers given the simplicity that it promises. To be clear, the results in Table 1 are very promising despite my criticism about using RMSE and of not conditioning on data. It is quite plausible that further work based on this paper would demonstrate better mixing across disconnected modes as claimed here.

[Author Response · NeurIPS 2019]

# NeurIPS 2019: Pseudo-Extended Markov chain Monte Carlo (paper ID: 2415)

We would like to thank the reviewers for dedicating their time to review our paper and the helpful feedback they have provided to improve the quality of this work. All of the reviewers' minor comments and corrections have been added to the paper. Below, we address the reviewers' main questions.

**Reviewer 1**

1. ***The paper focuses on HMC sampling. Do you believe that PE can be used in MH sampling or in SMC sampling? With discrete variables?*** The pseudo-extended approach modifies the target and not the sampler and so other sampling algorithms (e.g. MH and SMC) could be used to replace the original target distribution with the pseudo-extended target. As for discrete variables, sampling from discrete state-spaces can be challenging due to issues of dimensionality (e.g. Boltzmann example in 4.2). There is a similar motivation for concrete distributions. Unfortunately, HMC can't be applied in the discrete setting due to discontinuous gradients. However, transforming the problem into a continuous space (e.g. Boltzmann relaxation) means that we can use efficient gradient-based MCMC algorithms like HMC to explore the posterior space efficiently.

2. ***Could you provide intuition on what role $\pi(\beta)$ and $g(\beta)$ play, with respect to characteristics of the inference problem? From Fig. 2, $\beta$ plays a key role. How do you recommend setting $\pi$ and $g$ to best estimate $\beta$?*** This is a good question and something we have now expanded on in the paper. $\pi(\beta)$ is a prior on the tempering and in the experiments we assumed a uniform prior, however, in practice the user may have prior information about the multi-modality of the target, for example, whether the modes are far apart or tightly packed together, and so could choose a Beta prior which places more prior mass closer to 1 or 0, respectively. As for $g$, there's a lot of scope in future research to explore how this can be used to improve the mixing of the $\beta$ parameters. For example, something that we have recently experimented with is $g(\beta) = \mathcal{N}(0, \Sigma)$, where $\Sigma_{ij} > -1/(N-1)$ and $\Sigma_{ii} = 1$. This gives a positive definite matrix for $\Sigma$ where the $\beta$ pseudo-samples are negatively correlated and so repel each other allowing for better exploration of the $\beta$-space.

3. ***Could you please explain briefly how the models you wrote in Stan ensure that the sampler implements Algorithm 1 from Appendix B?*** Essentially, the Stan software applies HMC sampling to a specified log-target density function. Therefore, it's quite straightforward to implement pseudo-extended HMC within Stan by replacing the original log-target density with the pseudo-extended log-target density. Note that we could use Algorithm 1 directly for standard HMC (i.e. without Stan) and tune the step-size and length-scale parameters manually.

4. ***Looking through the Stan programs, the PE sampling code tends to be longer and more complicated. Have you thought about ways to reduce the complexity of the sampling code?*** From the Stan files you may notice that the pseudo-extended implementation is quite similar across each of the models. We should be able to simplify the code significantly by creating a "pseudo-extended" function at the top of the Stan file and call this function in place of the usual log-target, i.e. `target+=pseudo-extended()`. We will implement this in the version of the code that we release on Github.

**Reviewer 2**

1. ***As a minor comment in line 58, it would be good to state that delta is an arbitrary differentiable function.*** This is a good point and we've corrected this in the paper.

**Reviewer 3**

1. ***The experiments in 4.1 and 4.2 use the RMSE error of the target variables which is quite unusual. The more appropriate accuracy measure is the predictive likelihood of held out data vs inference time. Further, there is no conditioning on data in 4.1 and 4.2 so these are very trivial experiments in that all we are trying to do is generate a sample from the prior.*** The advantage of using the RMSE in 4.1 and 4.2 is that for these examples we can calculate the expectations exactly which provides us with ground-truth for comparison. We were also aiming to reflect what authors of previous works have also reported. We agree that predictive accuracy is important and we report this in 4.3. By considering both challenging simulated examples (4.1 and 4.2) and real-data scenarios (4.3) we can explore a wider range of posterior behaviour while also linking directly with previously considered models from the literature.

2. ***Report predictive log likelihood of held out data versus time. (For example see the Stan ADVI paper which has various graphs of this style).*** We've looked at the Stan ADVI paper and plots of this type will be replicated in our paper.

3. ***The paper should give some suggestions to tune the hyperparameter N.*** This is a good point and we've added some discussion on selecting $N$ in the manuscript.

[Meta-Review · NeurIPS 2019]

Reviewers reached consensus that the paper makes a valuable contribution for MCMC. There are specific suggestions for improving the experiments that we ask the authors to seriously consider.